# The Good, the Bad, and the Invisible with Its Opportunity Costs: Introduction to the 'J' Special Issue on "the Impact of Artificial Intelligence on Law"

Ugo Pagallo * and Massimo Durante

Department of Law, University of Turin, 10124 Torino, Italy; massimo.durante@unito.it
* Correspondence: ugo.pagallo@unito.it

**Abstract:** Scholars and institutions have been increasingly debating the moral and legal challenges of AI, together with the models of governance that should strike the balance between the opportunities and threats brought forth by AI, its 'good' and 'bad' facets. There are more than a hundred declarations on the ethics of AI and recent proposals for AI regulation, such as the European Commission's AI Act, have further multiplied the debate. Still, a normative challenge of AI is mostly overlooked, and regards the underuse, rather than the misuse or overuse, of AI from a legal viewpoint. From health care to environmental protection, from agriculture to transportation, there are many instances of how the whole set of benefits and promises of AI can be missed or exploited far below its full potential, and for the wrong reasons: business disincentives and greed among data keepers, bureaucracy and professional reluctance, or public distrust in the era of no-vax conspiracies theories. The opportunity costs that follow this technological underuse is almost terra incognita due to the 'invisibility' of the phenomenon, which includes the 'shadow prices' of economy. This introduction provides metrics for such assessment and relates this work to the development of new standards for the field. We must quantify how much it costs not to use AI systems for the wrong reasons.

**Keywords:** artificial intelligence (AI); legal governance; opportunity costs; standards; underuse of AI

## 1. Today's State-of-the-Art

Scholars and institutions have been increasingly debating the opportunities and threats brought forth by AI in recent years, as much as the impact of AI on tenets of the law [1]. The opportunities should be grasped in accordance with the manifold uses of AI systems that are arguably 'good' for moral, social, economic, and legal reasons [2]. The use of AI for diagnostics and prevention, precision medicine and medical research, clinical decision making and mobile health, healthcare management and service delivery offers a sound example of the good use of AI [3,4]. Further instances include the protection of the environment and current climate crisis [5,6]. AI systems can help us to reduce pollution emissions (e.g., monitoring and intelligent management of networks and consumption); figure out better ways to prevent natural disasters; make a smart use of resources, such as water and electricity; or strengthen the circular economy through, e.g., the monitoring and predictive management of the waste cycle [7]. In the words of the European Commission, "data, combined with digital infrastructure (e.g., supercomputers, cloud, ultra-fast networks) and artificial intelligence solutions, can facilitate evidence-based decisions and expand the capacity to understand and tackle environmental challenges" [8].

There is the other side of the coin, however. The list of opportunities and good uses of AI should be complemented with its threats and constraints. One of the first fields in which scholars and institutions have discussed the 'bad' uses of AI is the military sector with the use of autonomous lethal weapons as far back as the mid-2000s [9]. Further bad uses of AI have been examined by criminal lawyers [10], or experts in the fields of healthcare and environmental sustainability [11], agriculture [12], and transportation [13]. For example,

as regards the health sector, AI systems may trigger tricky issues of transparency and bias, privacy and security, fairness and equity, data quality and data aggregation, in hospitals, wards, or clinics [14]. Likewise, AI is likely to add growing concerns for the increasing volume of e-waste and the pressure on rare-earth elements generated by the computing industry [15]. Advanced AI technologies often require massive computational resources that hinge on large computing centers and these facilities have a very high energy requirement and carbon footprint [16]. Considering such different 'bad uses' of AI, it is then unsurprising that a hundred declarations on the ethics of AI have piled up over the past few years [14]. In addition, recent proposals for AI regulation, such as the European Commission's AI Act, have further ignited the debate among scholars on how to discipline the 'good' and 'bad' uses of AI [17].

Still, by stressing the good and bad uses of AI, our intent is not to suggest that AI, or any other technology for that matter, is simply neutral, i.e., a bare means to attain whatsoever end. By disclosing a new set of opportunities and constraints, AI affects how we understand ourselves and our environment. AI realigns traditional issues of rights and duties, autonomy and accountability, negligence, and duties of care [18]. To address this mix of good and bad uses of technology, scholars have increasingly focused on further issues of legal governance [19–22]. One of the most relevant conclusions of this debate is that traditional top-down regulatory approaches fall short in tackling the normative challenges of AI. At the institutional level, it is noteworthy that EU law has endorsed a co-regulatory model of legal governance in the fields of data protection, communication law, and data governance, i.e., the 2020 proposal of the Commission for a new Data Governance Act. Scholars sum up this institutional work in progress also, but not only, in EU law, with different formulas of legal governance, such as co-regulation [23], supervised self-regulation [24], binary governance [25], and more [26]. This lexical discord reflects the complexity of the field and its dynamics. The long list of initiatives and proposals discussed at the EU level—from the *Digital Services* and *Digital Markets Act* from December 2020 to the *Cybersecurity Act* from July 2021, in addition to the troubles with the GDPR, the Data Governance Act, the AI Act, or the initiatives for a Green Deal—illustrates the panoply of open problems with which we are dealing, and why further work on the legal governance of AI is necessary.

Among the bad uses of AI and the corresponding problems of legal governance, special attention should be drawn to the distinction between the misuse, overuse, and underuse of AI [2]. Since the focus of scholars on the bad uses of AI has been mostly concerned with the possible misuses and overuses of the technology, problems of its underuse have been overlooked. It is our contention that legal scholars should start paying attention to the opportunities provided by AI that can be missed or exploited far below its full potential, because of wrong reasons. This scenario was partially explored by moral philosophers, economists, and institutions. For example, in a press release of the European Parliament, in September 2020, it was suggested that the "underuse of AI is considered as a major threat" [27]. Likewise, a similar problem was stressed by the 2019 document of the G20 in Tokyo and the 2021 Final Report of the US National Security Commission on Artificial Intelligence [28]. Whereas such authorities admit that the underuse of AI is a problem, one that is far from being solved, few legal scholars have examined why the attempts to tackle the "major threat" of AI underuse have failed. The legal issue appears, so to speak, to be 'invisible' because it is mostly ignored by today's declarations on the ethical principles of AI, or by current discussions on how to legally regulate and govern AI.

To explore this terra incognita of the underuse of AI, this introduction illustrates, first, the wrong reasons as to why AI systems are not employed in many cases, or are employed far below their full potential. These drivers of AI underuse include social norms and the forces of the market, and are often in competition with the regulatory claims of the law. AI may be underused due to bureaucracy, public distrust, bad journalism, fake news, the greed of both public and private data keepers, lack of infrastructure, and more. Therefore, how should lawmakers tackle such drivers of technological underuse?

## 2. The Underuse of AI

The law has traditionally dealt with cases of technological underuse on individual basis and in connection with notions of diligence and duties of care in contracts and extra-contractual liability, under the umbrella of 'current state-of-the-art'. Personal and corporate responsibility for the underuse of technological devices and systems has been examined in health law, e.g., national health services, cybersecurity law, communication law, and data protection. However, what is in this paper under scrutiny with the underuse of AI systems is rather different. In addition to personal and corporate responsibility, the focus should be on the duties of national governments and international organizations. There is a growing number of AI systems that are employed far below their full potential in medicine [29], healthcare [30], sustainable development [5], environmental protection [7], circular economy [31], agriculture [32], and more. The scale of the threat is such that, remarkably, most policy makers and legislators, at both national and international levels, have endorsed a pro-active approach to these challenges of AI. According to Article 3.2(a) of the G20 document from 2019, a proactive approach means that "governments should promote a policy environment that supports an agile transition from the research and development stage to the deployment and operation stage for trustworthy AI systems. To this effect, they should consider using experimentation to provide a controlled environment in which AI systems can be tested, and scaled-up, as appropriate" [33].

We (as have some other scholars) have recommended this 'experimental approach' over the past years [34,35], and we are thus glad that some of our recommendations have been adopted: an European Artificial Intelligence Board, set up with Art. 56 of the AI Act [2,21]; the new logs duties of Art. 12 on record keeping [36]; the set-up of regulatory sandboxes of Art. 53 [2,37]; etc.

However, it seems fair to concede that the G20s 'agile transition' from labs and research centers to society remains a difficult task, especially considering that, despite the general framework provided by the AI Act, the risk of underuse is highly context dependent and needs congruent context-dependent policies [38]. What is underused in, say, the health sector often entails quite different problems than issues of AI underuse in public transport, sustainable development, circular economy, agriculture, etc. Institutions and lawmakers have thus to address the threats of technological underuse considering the specificities of each field with ad hoc regulatory proposals. This has been so far the strategy of the European Commission. In addition to 'horizontal interventions'—such as the GDPR or the AI Act—the Commission has promoted 'vertical initiatives' as the e-health stakeholder group initiative [39], the Common European Green Deal data space on AI and environmental law [40], or the Data Governance Act on interoperability and sharing of data [41]. This is not to say that most problems related to the underuse of AI depend on EU law. On the contrary, they mostly depend on national choices and corresponding jurisdictions, whereas, in most legal systems of EU member states, there is no such a thing as an enforceable obligation upon national governments to proactively tackle the 'major threat' of AI underuse.

For example, in health law, states are committed to guarantee at the international level the individual right to health enshrined in Art. 25 of the 1948 Universal Declaration of Human Rights. Still, in accordance with Art. 3(4) of the international health regulations, states have "the sovereign right to legislate and to implement legislation in pursuance of their health policies". The same holds true for cases of AI underuse for environmental purposes: in EU law, for instance, the Court of Justice has granted each EU institution and all member states, "a wide discretion regarding the measures it chooses to adopt in order to implement the environmental policy" [42]. Therefore, we can expect not only different national strategies, if any, concerning the 'major threat' of the underuse of AI, but such strategies will necessarily be related to the specificities of each field, that is, the underuse of AI in the health sector, in environmental law, for agriculture, in public transport systems, etc. [43–46].

Regardless of the field (and jurisdiction) under scrutiny, all lawmakers and policy makers must however face some common problems. The threat of AI underuse may in fact depend on legal regulations and case law of the courts that trigger further cases of technological underuse with their provisions, including those which aim to tackle threats of underuse, e.g., Art. 53 of the European Commission's proposal for a new AI Act. In addition, further regulatory systems, such as the forces of the market or of social norms, can induce the underuse of technology [47]. In the words of the European Parliament, "underuse could derive from public and business' mistrust in AI, poor infrastructure, lack of initiative, low investments, or, since AI's machine learning is dependent on data, from fragmented digital markets" [27]. Consequently, the problem revolves around how the law should tackle such drivers of technological underuse, in particular, the wrong reasons as to why AI systems are often used far below their full potential because of business disincentives and greed among data keepers, bureaucracy and professional reluctance, or public distrust in the era of no-vax conspiracy theories [48].

Over the past few years, many legal systems have adopted a mix of soft law and mechanisms of coordination and cooperation to tackle such hurdles. We mentioned the co-regulatory approach of EU lawmakers with its variants in the e-health sector, environmental protection, the Green Deal, or the governance of current data-driven societies. Further examples of other jurisdictions are similarly context dependent [38]. A mix of soft law and coordination mechanisms have been set up from Australia to the U.K., from Singapore to the U.S.A., to prevent or tackle cases of AI underuse in medicine and health. For instance, according to the "Stakeholder Engagement Framework" of the Department of Health of the Australian Government [49], five principles of engagement should help us to tackle cases of technological underuse, through a clear understanding of the aims of the engagement, its inclusiveness, timeliness, transparency, and respectfulness, as regards the expertise, perspective, and needs of all stakeholders [50]. Traditional top-down approaches of legislation are thus complemented with five levels of engagement, from simple information to consultation, involvement, collaboration, and finally, delegation of legal powers to stakeholders [49].

We mentioned current discussions on models of legal governance for AI and the fact that national and international institutions, from the European Parliament to the G20 [27,33], the OECD [51], or ITU and the WHO [52], have insisted time and again on issues of technological underuse. Current discussions on models of co-regulation, supervised self-regulation, binary governance, etc. should thus help us to understand why most attempts of legislators to solve problems of technological underuse have failed. Our conjecture is that the problem does not depend on the plan of legislators and policy makers to complement traditional top-down approaches of regulation with coordination mechanisms and methods of cooperation and co-regulation. Rather, our contention is that such mechanisms of coordination, engagement, collaboration, etc. have piled up with no methods through these institutional proposals and initiatives. Seminal research on the underuse of AI in the health sector seems to confirm this view [53].

In addition to the much-debated issues of legal governance, the underuse of AI raises, however, some problems of its own. Regardless of the field that is under scrutiny, the underuse of something or someone entails what economists dub as 'opportunity costs'. Such costs include lower standards in products and services, the redundancy or inefficiency of such services, much as the 'shadow prices' of the economy [54]. There have been many attempts to quantify such opportunity costs in traditional fields of research that cover national health services, transportation systems, and their cost analysis. For example, in the U.S.A., the opportunity costs of ambulatory medical care are esteemed around 15%: "For every dollar spent in visit reimbursement, an additional 15 cents were spent in opportunity costs" [55]. In the U.K., the opportunity costs of the *National Health Service* were esteemed around GBP 10 million each year, from 2002–2003 to 2012–2013, although such figures seem to underestimate the phenomenon [56]. Further research on opportunity costs regards

thresholds for cost-effectiveness analysis [57], the development of value frameworks for funding decisions [58,59], and more.

As far as we know, however, there have been few attempts to understand—and quantify—the 'opportunity costs' triggered by the underuse of AI in any sector. The reasons for this silence, which adds to the invisibility of the phenomenon, are many: the novelty of the issue, the difficulty of the task, and the fact that the 'major threat' of AI underuse has simply been overlooked with its opportunity costs. Therefore, we need to fill this gap in current analyses of AI. The next section aims to provide some guide for this further facet of our terra incognita.

## 3. The Opportunity Costs of AI

The quantification of the opportunity costs that follow the underuse of AI in today's societies is, admittedly, no easy task. The difficulty of the assessment hinges on the traditional hurdles of econometrics and the novelty of the phenomenon. After all, many fields of the law and AI lack standards and metrics not only for the assessment of the opportunity costs of AI, but also for the costs of its overuse or misuse. In environmental law, for example, it is still an open issue how we should determine the footprint of AI, considering its energy costs, or carbon emissions [60], but also the metrics AI systems are optimized for, or further efficiency metrics for AI, as model training [61]. In some other fields, as health law, there is a certain consensus on such indexes, as the health-related quality of life (HRQOL), or the quality-adjusted life-year (QALF), and yet, it is still far from clear the amount of the opportunity costs for the underuse of AI in medicine and health care [62].

In addition to the different parameters and metrics to be taken into account, such as the assessment of resources that have no market vis à vis methods for value-based pricing, attention should be drawn to further technicalities of the analysis on the opportunity costs, such as their 'incremental ratio' [63], and the speed of technological innovation that accelerates cases of underuse [64]. We expect striking differences among jurisdictions, and between different sectors, e.g., the opportunity costs of the health sector vis à vis the opportunity costs of environmental protection in, say, Germany and Spain. This conjecture suggests that no single answer exists for the opportunity costs that follow 'the' underuse of AI, since such opportunity costs depend on the field and types of AI systems under scrutiny, as much as on the countries and jurisdictions that are examined. Several successful stories of AI, for example in medicine and health care, come from low- or medium-income countries, casting further light on the underuses of AI in high-income countries. There are AI systems that predict birth asphyxia in children by scrutinising the birth cry of a child via mobile phones in Nigeria [65]; AI apps that offer guidance and recommendations to nurses and paramedic personnel in India and sub-Saharan Africa [66]; and AI systems that detect water contamination [67], control dengue fever transmission [68], or Ebola outbreaks [69].

From this latter viewpoint, therefore, the underuse of AI can be grasped as a sort of paradox of the wealthy, a waste of time, money, resources, and quality of life. However, it should also be noted that many of the current initiatives in high-income countries and scholarly debate on how to ameliorate such initiatives could properly be extended to the context of low- and medium-income countries [21]. The scalability and modularity of the co-regulatory models adopted by many Western institutions can in fact tackle issues of technological underuse in developing countries, for example, fleshing out which specific threats of AI underuse should be prioritized through initiatives that can be scaled up with the modularization of the projects. This approach also fits high-income countries that have no experience of co-regulatory approaches, for example, Italy [53].

Another reason why the underuse of AI will increasingly attract the attention of scholars has to do with the growing economic relevance of such underuse. Although quantifying the opportunity costs of AI can be tricky, the problem is not 'whether' but 'how much.' For example, working on the problem of AI underuse in the health sector, one of the authors of this paper esteems that in some countries, e.g., Italy, which invests around

9% of its gross domestic product (GDP) in the public health sector, the opportunity costs of AI underuse for health and care may amount to 1% up to 2% of the Italian GDP [53]. The appraisal includes the optimization of services, quality of standards, and the 'shadow prices' for people's useless waiting lists and movements, traffic congestion, pollution, etc. More than 90% of people arriving in emergency wards in public hospitals in Italy should have stayed at home [53]. We expect interesting comparisons with other legal systems and traditions, as regards the opportunity costs of their health sector, their strategy for public transport and sustainable development, agriculture, circular economy, and more.

Admittedly, the evaluation of the opportunity costs is not a simple illustration of reality. The problem to determine such costs partially overlaps with a crucial aspect of technological regulation, such as the development of standards and metrics [20]. They may regard duties of transparency [70], data disclosure [71], or help for align in metrics [61,72]. Whereas the lack of standards often represents a powerful source of technological underuse, e.g., lack of standards for data accuracy and interoperability [73], work on the underuse of AI can help us to attain such legal standards through the evaluation of that which obstructs the employment of AI, e.g., lack of standards on common metrics. We already insisted on the current activism of lawmakers and how their proposals, e.g., the AI Act, could be amended and implemented also, but not only, as regards the development of new "harmonized standards" pursuant to Art. 40 of the Act, and the top-down approach to standards enshrined in Art. 41. Since most international institutions and some lawmakers, such as the European Parliament, concede that the threat of AI underuse is far from solved, we expect further initiatives of policy makers and legislators, and the corresponding debate of scholars on how to ameliorate such initiatives on, e.g., metrics and standards for (determining the opportunity costs of) AI systems.

A final reason why the underuse of AI will increasingly attract the attention of scholars concerns the exponential advancement of technology. No sci-fi scenario of super artificial intelligences is needed [1] to admit that the speed and scale of AI innovation will progressively make that which is not, i.e., the underuse of AI, more and more 'visible.' The analysis has mentioned several possible uses of AI in medicine and health care, agriculture and transportation, environmental protection and circular economy, most of which are piling up, however, in labs and research centres. Our contention is that the scale and speed of AI innovation will increase the perception of that which is not used, or is used far below its full potential, by increasingly drawing the attention of the media, of scholars working on the normative challenges of AI, and the public at large. This is not the first time that advancements of technology make some of its underuse apparent, for example, the adoption of full colour television screens in the late 1970s in Italy, whereas the technology was available in that country since the early 1960s. The speed of AI innovation is exponential, and we may thus wonder whether lawmakers and international institutions are prepared to tackle such a challenge.

## 4. This Special Issue

The 'Law and AI' is an extremely dynamic field that requires constant reviewing and update because of the speed of technological innovation and the current activism of legislators and policy makers. We already singled out four areas in which this constant reviewing and updating appears critical: (i) the impact of AI on tenets of the law, e.g., personhood, and forms of legal reasoning; (ii) the legal regulation of 'bad' uses of AI; (iii) the legal governance of AI; and, (iv) how the law relates to the norms of further regulatory systems, e.g., social norms or the forces of the market, as drivers, for example, of the underuse of AI in medicine and health care.

Correspondingly, the contributions of this Special Issue on the legal impact of AI can be presented in accordance with such four areas of philosophical reflection and constant update:

(i) The papers of Woodrow Barfield [74], Lance Jaynes [75], and Billi et al. [76] on the impact of AI on tenets of the law;

(ii)    The papers of Mateja Durovic and Jonathon Watson [77], and Martin Ebers et al. [78] on new legal regulations for AI;

(iii)   The papers of Pompeu Casanovas et al. [79], and Rolf Weber [80] on models of legal governance;

(iv)   The papers of Francesco Sovrano et al. [81] and Daniel Trusilo and Thomas Burri [82] on how the law relates to the norms of further regulatory systems, as in the case of ethics.

In particular, 'A Systems and Control Theory Approach for Law and Artificial Intelligence' deals with how the law should govern the use of algorithmically based systems [74]. Most of the time, scholars focus on the external behavior of AI systems. Barfield draws our attention to the legal relevance of the feedback loop and other control variables that relate the systems' input to the desired output.

'The Question of Algorithmic Personhood and Being' dwells on a classic topic of the law and AI, i.e., personhood, by examining the increasing ability for virtual avatars to engage and interact with our physical world [75]. Jaynes proposes a field-wide shift, including Japanese inspiration for our mostly Western reflections, to address the pressing issues brought about by such virtual avatars.

The paper 'Argumentation and Defeasible Reasoning in the Law' scrutinizes current logic-based approaches to defeasible reasoning, considering the logical model (knowledge representation), the method (computational mechanisms), and the technology (available software resources). Billi et al. illustrate the benefits of an argumentation approach to most real-world implementations of AI systems in the legal domain [76].

'Nothing to Be Happy about: Consumer Emotions and AI' is concerned with the technologies that are able to detect and to respond to individual emotions [77]. Durovic and Watson assess current regulations of data protection and consumer law, to stress that current legal protection of emotions is mostly non-existent, and the AI Act simply overlooks this threat.

A critical assessment of the whole AI Act has been carried out by Martin Ebers and other members of the Robotics and AI Law Society (RAILS) [78]. Although the design of the Act and its risk-based regulatory approach seem robust, there are flaws and parts of the proposal should be clarified, or improved, starting with the definition of AI, down to the role that self-regulation and internal controls by AI providers should play in the new EU regulation.

'Law and Socio-legal Governance, IOT, and Industry 4.0' delves deeper into issues of legal governance in digital and blockchain environments, illustrating how the rule of law should be implemented in such new contexts [79]. Casanovas et al. aim to validate legal information flows and hybrid agents' behavior through a phenomenological and historical approach to legal and political forms of governance that distinguish 'enabling' from 'driving' regulatory systems.

'Artificial Intelligence ante portas: Reactions of Law' similarly insists on how AI and algorithmic decision making impact fundamental normative principles, such as non-discrimination, human rights, transparency, etc. [80]. Weber stresses the risk that top-down regulations, such as the AIA, may hinder innovation, therefore proposing a more flexible and innovation-friendly combination of regulatory models in the legal domain.

'Metrics, Explainability and the European AI Act Proposal' draws the attention to the standardization process ongoing in EU law [81]. Sovrano et al. focus on specific explicability obligations and metrics for measuring the degree of compliance, envisaging the legal and ethical requirements that such metrics should possess, to implement them in a practical way.

Last, but not least, 'The Ethical Assessment of Autonomous Systems in Practice' examines the normative challenges arising in the field of autonomous robotic systems, e.g., the four-legged robotic system designed to compete in the U.S. Defense Advanced Research Projects Agency (DARPA) subterranean challenge [82]. Trusilo and Burri illustrate

how the law relates to the norms of further regulatory systems, through the instance of a multi-modal autonomous search operation in the field of applied ethics.

Drawing on these nine papers, the time is ripe for the conclusions of this introduction.

## 5. Conclusions

Mimicking the exponential growth of AI, work on the legal impact of AI has grown exponentially over the past few years. Among the reasons for this popularity, we noted the uniqueness of AI [2], the speed of its advancement [64], and current activism of lawmakers, which requires constant reviewing and updating to keep up with today's debate in the field of the law and AI [1,83].

This Special Issue includes nine papers that focus on the main areas in which this constant reviewing and update appears critical, that is, (i) the impact of AI on tenets of the law, such as personality and legal reasoning; (ii) the regulation of 'bad' uses of AI, as occurs with the AI Act; (iii) the legal governance of AI and current discussions on regulatory models for both 'good' and 'bad' uses of AI; and (iv) how the law relates to the norms of further regulatory systems, as the forces of the market or of social mores. This Special Issue presents analyses that, therefore, are original for four reasons. Either they cast new light on classic discussions on legal reasoning and law and philosophy [74–76]; or they address the novelties of current legal frameworks and initiatives of policy makers [77,78]; or they further develop models of governance that should fit the challenges and opportunities brought forth by AI [79,80]; or they inspect how the interaction between different regulatory systems, e.g., ethics and the law, evolves due to the dynamics of technological innovation [81,82]. In addition, this introduction has insisted on problems that are either new because they are unprecedented, or new because they are mostly overlooked by scholars.

Such different reasons for novelty can of course overlap. In our analysis on the underuse of AI, for example, the focus was on (i) some new legal proposals of AI regulation; (ii) the problems of legal governance for AI; and (iii) the role that multiple regulatory systems play vis à vis the regulatory aims of the law. By insisting on the threat of AI underuse, attention was thus drawn to a problem mostly overlooked by scholars that, nevertheless, will be more and more evident due to the incremental advancements of technology and the corresponding growth of the opportunity costs that follow such underuse. Although there is strong evidence that AI systems are already wrongly not used, or used far below their full potential in healthcare, environmental protection, etc., we are in terra incognita, because current discussions on the parameters, according to which we should evaluate and quantify such underuse of AI with its opportunity costs, are in their infancy.

This paper explored the facets of this terra incognita, providing some guide on why most of the attempts of lawmakers have fallen short in dealing with the 'threat of AI underuse', and how those attempts can be ameliorated in terms of governance and through the development of new standards and metrics to determine the opportunity costs of AI. The law can indeed be both the cause and the solution to this crucial challenge of today's societies.

**Author Contributions:** Conceptualization: U.P. and M.D.; Methodology: U.P. and M.D.; Investigation: U.P. and M.D.; Resources: U.P: and M.D.; Writing—original draft preparation: U.P. and M.D.; Writing—review and editing: U.P. and M.D. All authors have read and agreed to the published version of the manuscript.

**Funding:** This research received no external funding.

**Conflicts of Interest:** The authors declare no conflict of interest.

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
