# Peer review of "The Good, the Bad, and the Invisible with Its Opportunity Costs: Introduction to the ‘J’ Special Issue on “the Impact of Artificial Intelligence on Law”"

_2571-8800, doi:10.3390/j5010011_

Round 1

Reviewer 1 Report

-The paper presents opportunity costs related to the impact of Artificial Intelligence on Law.
-I think the authors should have mentioned more about the technological infrastructure required for an adequate adoption of Artificial Intelligence in the different areas mentioned. Sometimes it is not just the current laws or conventions that need to be updated but also the technological infrastructure, such as 5G for Internet of Things.
-Be consistent when using names, for example in the case of "UK" and "U.K.".
-I think the paper presents a good introduction to the Special Issue.

Author Response

Thanks to the reviewers for their valuable comments.

As regards the first reviewer and his/her remark on the role that ‘technological infrastructure’ plays for an ‘adequate adoption of AI’, we agree with this remark and, therefore, we have added a further reference to this issue at the end of the first section. We return to this in section 2, reference no 27. And thanks for drawing our attention to the consistency of some acronyms, e.g., UK vs U.K. We uniformed the style throughout the paper.

As regards the second reviewer and his/her remark on cutting some parts of the paper which may be redundant, we doublechecked the text and, apart from some light corrections, decided to let most of the paper unmodified, since most information is essential and the paper is not so long. However, we incorporated two of the papers recommended by the reviewer that help strengthening our analysis. Thank you! You’ll find them as references no. 44 and 45.

All the best,

Ugo Pagallo & Massimo Durante

Reviewer 2 Report

Dear Authors,

many thanks for your introduction. It faces a really hot topic and it well written. My only suggestion is to cut some parts which may be redundant, as the paper overall is quite (too much?) long.

Also, I suggest to consider also the following paper in your analysis:

  • DOI: 10.1007/s13244-018-0645-y
  • DOI: 10.1016/S2589-7500(20)30292-2
  • DOI: 10.3389/frobt.2021.789327
  • DOI: 10.3390/radiation1040022

    Author Response

    Thanks to the reviewers for their valuable comments.

    As regards the first reviewer and his/her remark on the role that ‘technological infrastructure’ plays for an ‘adequate adoption of AI’, we agree with this remark and, therefore, we have added a further reference to this issue at the end of the first section. We return to this in section 2, reference no 27. And thanks for drawing our attention to the consistency of some acronyms, e.g., UK vs U.K. We uniformed the style throughout the paper.

    As regards the second reviewer and his/her remark on cutting some parts of the paper which may be redundant, we doublechecked the text and, apart from some light corrections, decided to let most of the paper unmodified, since most information is essential and the paper is not so long. However, we incorporated two of the papers recommended by the reviewer that help strengthening our analysis. Thank you! You’ll find them as references no. 44 and 45.

    All the best,

    Ugo Pagallo & Massimo Durante
